# A Variational Autoencoder for Probabilistic Non-Negative Matrix Factorisation

## Abstract

We introduce and demonstrate the variational autoencoder (VAE) for probabilistic non-negative matrix factorisation (PAE-NMF). We design a network which can perform non-negative matrix factorisation (NMF) and add in aspects of a VAE to make the coefficients of the latent space probabilistic. By restricting the weights in the final layer of the network to be non-negative and using the non-negative Weibull distribution we produce a probabilistic form of NMF which allows us to generate new data and find a probability distribution that effectively links the latent and input variables. We demonstrate the effectiveness of PAE-NMF on three heterogeneous datasets: images, financial time series and genomic.

## 1 Introduction

### 1.1 Non-Negative Matrix Factorization

There has been a considerable increase in interest in NMF since the publication of a seminal work by Lee & Seung (1999) (although earlier Paatero & Tapper (1994) had studied this field) in part because NMF tends to produce a sparse and parts based representation of the data. This sparse and parts based representation is in contrast to other dimensionality reduction techniques such as principal component analysis which tends to produce a holistic representation. The parts should represent features of the data, therefore NMF can produce a representation of the data by the addition of extracted features. This representation may be considerably more interpretable than more holistic approaches.

Consider a data matrix $\mathbf{V} \in \mathbb{R}^{m \times n}$ with $m$ dimensions and $n$ data points which has only non-negative elements. If we define two matrices, also with only non-negative elements: $\mathbf{W} \in \mathbb{R}^{m \times r}$ and $\mathbf{H} \in \mathbb{R}^{r \times n}$, then non-negative matrix factorisation (NMF) can reduce the dimensionality of $\mathbf{V}$ through the approximation $\mathbf{V} \approx \mathbf{WH}$ where, generally, $r < \min(m, n)$.

The columns of $\mathbf{W}$ make up the new basis directions of the dimensions we are projecting onto. Each column of $\mathbf{H}$ represents the coefficients of each data point in this new subspace. There are a range of algorithms to conduct NMF, most of them involving minimising an objective function such as

$$\min_{\mathbf{W},\mathbf{H}} ||\mathbf{V} - \mathbf{WH}||_{\mathrm{F}}^2 \text{ subject to } W_{i,j} \geq 0, H_{i,j} \geq 0.$$

### 1.2 Using Autoencoders for NMF

Several authors have studied the addition of extra constaints to an autoencoder to perform NMF (Lemme et al., 2012; Ayinde et al., 2016; Hosseini-Asl et al., 2016; Smaragdis & Venkataramani, 2017). These methods show some potential advantages over standard NMF including the implicit creation of the $\mathbf{H}$ matrix and straightforward adaptation to online techniques.

In Figure 1 we show a representation of a one-hidden layer autoencoder for performing NMF. We feed in a data-point $\mathbf{v} \in \mathbb{R}^{m \times 1}$, the latent representation is produced by $\mathbf{h} = f(\mathbf{W}_1 \mathbf{v})$ where $f$ is an element-wise (linear or non-linear) function with non-negative ouputs, $\mathbf{h} \in \mathbb{R}^{r \times 1}$ is the representation in the latent space and $\mathbf{W}_1 \in \mathbb{R}^{r \times m}$ contains the weights of the first layer. It is also possible to add additional layers to make a deeper network with multiple hidden layers before arriving at the constriction layer which produces the $\mathbf{h}$ output. The final set of weights must be kept non-negative

with an identity activation function such that $\mathbf{x} = \mathbf{W}_f \mathbf{h}$ where $\mathbf{W}_f \in \mathbb{R}^{m \times r}$. The final weights of the autoencoder then can be interpreted as the dimensions of the new subspace with the elements of $\mathbf{h}$ as the coefficients in that subspace. The network is then trained so that $\mathbf{x} \approx \mathbf{v}$.

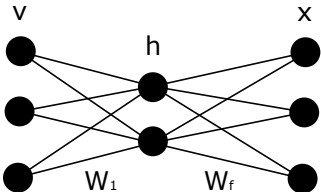

Figure 1: Diagram of an autoencoder designed to perform NMF. The weights of the final layer, $\mathbf{W}_f$, become the directions of the subspace, with the outputs of the hidden layer, $\mathbf{h}$, as the coefficients in that new subspace. The activation function that produces $\mathbf{h}$ must be non-negative as must all the elements of $\mathbf{W}_f$.

## 1.3    VARIATIONAL AUTOENCODERS

NMF and autoencoders both produce a lower dimensional representation of some input data. However, neither produce a probability model, just a deterministic mapping. It is also not obvious how to generate new data from these lower dimensional spaces. Generative models solve both of these problems, enabling a probability distribution to be found linking the input and latent spaces whilst enabling new data to be created.

One of the most popular recent generative model is the variational autoencoder (VAE) (Kingma & Welling, 2013; Rezende et al., 2014). A VAE is a probabilistic model which utilises the autoencoder framework of a neural network to find the probabilistic mappings from the input to the latent layers and on to the output layer. Unlike a standard autoencoder the VAE finds a distribution between the latent and seen variables, which also enables the production of new data by sampling from the latent distributions.

## 1.4    PROBABILISTIC NON-NEGATIVE MATRIX FACTORISATION

Several authors have presented versions of NMF with a probabilistic element (Paisley et al., 2014; Cegmil, 2008; Schmidt et al., 2009) which involve the use of various sampling techniques to estimate posteriors. Other work has been done using hidden Markov models to produce a probabilistic NMF (Mohammadiha et al., 2013) and utilising probabilistic NMF to perform topic modelling (Luo et al., 2017). However, to our knowledge no one has utilised the ideas behind the VAE to perform NMF.

Although probabilistic methods for NMF have been developed, even a full Bayesian framework still faces the problem that, for the vast majority of problems where NMF is used, we have little idea about what is the appropriate prior. We would therefore be forced to do model selection or introduce hyperparameters and perform inference (maximum likelihood or Bayesian) based on the evidence. However, as the posterior in such cases is unlikely to be analytic this is likely to involve highly time consuming Monte Carlo. In doing so we would expect to get results close to those we obtain using PAE-NMF. However, for machine learning algorithms to be of value they must be practical. Our approach, following a minimum description length methodology, provides a principled method for achieving automatic regularisation. Because it fits within the framework of deep learning it is relatively straightforward and quick to implement (using software such as Keras or PyTorch with built-in automatic differentiation, fast gradient descent algorithms, and GPU support). In addition, our approach provides a considerable degree of flexibility (e.g. in continuous updating or including exogenous data), which we believe might be much more complicated to achieve in a fully probabilistic approach.

The next section lays out the key alterations needed to a VAE to allow it to perform NMF and is our main contribution in this paper.

## 2 PAE-NMF

The model proposed in this paper provides advantages both to VAEs and to NMF. For VAEs, by forcing a non-negative latent space we inherit many of the beneficial properties of NMF; namely we find representations that tend to be sparse and often capture a parts based representation of the objects being represented. For NMF we introduce a probabilisitic representation of the vectors $\mathbf{h}$ which models the uncertainty in the parameters of the model due to the limited data.

### 2.1 IDEAS BEHIND PAE-NMF

Kingma & Welling (2013) proposed the VAE, their aim is to perform inference where the latent variables have intractable posteriors and the data-sets are too large to easily manage. Two of their contributions are showing that the "reparameterization trick" allows for use of standard stochastic gradient descent methods through the autoencoder and that the intractable posterior can be estimated.

In a standard autoencoder we take some data point $\mathbf{v} \in \mathbb{R}^m$ and run it through a neural network to produce a latent variable $\mathbf{h} = f(\mathbf{v}) \in \mathbb{R}^r$ where $r < m$ and $f$ is some non-linear element-wise function produced by the neural network. This is the encoding part of the network. We then run $\mathbf{h}$ through another neural network to produce an output $\hat{\mathbf{v}} = g(\mathbf{h}) \in \mathbb{R}^m$, which is the decoding part of the network. The hope is that due to $r < m$ the latent variables will contain real structure from the data.

A standard VAE differs in that instead of encoding a deterministic variable $h$ we find a mean, $\mu$, and variance, $\sigma^2$ of a Gaussian distribution. If we want to generate new data we can then sample from this distribution to get $h$ and then run $h$ through the decoding part of the network to produce our new generated data.

PAE-NMF utilises the same objective function as a standard VAE, so for each data-point, we are minimising:

$$\text{obj} = \mathbb{E}_{q_\phi(\mathbf{h}|\mathbf{v})} \left( -\log \left( p_\theta(\mathbf{v}|\mathbf{h}) \right) \right) + \text{D}_{\text{KL}}(q_\phi(\mathbf{h}|\mathbf{v})||p(\mathbf{h})) \tag{1}$$

$$\approx \frac{1}{2\sigma^2} ||\mathbf{v} - \hat{\mathbf{v}}||^2 + \text{D}_{\text{KL}}(q_\phi(\mathbf{h}|\mathbf{v})||p(\mathbf{h})) \tag{2}$$

where $\text{D}_{\text{KL}}$ is the KL divergence, $\phi$ and $\theta$ represent the parameters of the encoder and decoder, respectively, $\mathbf{v}$ is an input vector with $\hat{\mathbf{v}}$ the reconstructed vector, created by sampling from $p(\mathbf{h}|\mathbf{v})$ and running the $\mathbf{h}$ produced through the decoding part of the network. The first term represents the reconstruction error between the original and recreated data-point, which we assume to be approximately Gaussian. The second term is the KL divergence between our prior expectation, $p(\mathbf{h})$, of the distribution of $\mathbf{h}$ and the representation created by the encoding part of our neural network, $q_\phi(\mathbf{h}|\mathbf{x})$. We can interpret this as a regularisation term, it will prevent much of the probability density being located far from the origin, assuming we select a sensible prior. Another way of looking at it is that it will prevent the distributions for each datapoint being very different from one another as they are forced to remain close to the prior distribution.

This objective function can also be interpreted as the amount of information needed to communicate the data (Kingma et al., 2014). We can think of vectors in latent space as code words. Thus to communicate our data, we can send a code word and an error term (as the errors fall in a more concentrated distribution than the original message they can be communicated more accurately). The log-probability term can be interpreted as the message length for the errors. By associating a probability with the code words we reduce the length of the message needed to communicate the latent variables (intuitively we can think of sending the latent variables with a smaller number of significant figures). The KL divergence (or relative entropy) measures the length of code to communicate a latent variable with probability distribution $q_\phi(\mathbf{h}|\mathbf{v})$. Thus by minimising the objective function we learn a model that extracts all the useful information from the data (i.e. information that allows the data to be compressed), but will not over-fit the data.

## 2.2 STRUCTURE OF THE PAE-NMF

The structure of our PAE-NMF is given in Figure 2. The input is fed through an encoding neural network which produces two vectors $\mathbf{k}$ and $\boldsymbol{\lambda}$, which are the parameters of the $q_\phi(\mathbf{h}|\mathbf{x})$ distribution. The latent vector for that data-point, $\mathbf{h}$, is then created by sampling from that distribution. However, this causes a problem when training the network because backpropagation requires the ability to differentiate through the entire network. In other words, during backpropagation we need to be able to find $\frac{\partial h_i}{\partial k_i}$ and $\frac{\partial h_i}{\partial \lambda_i}$, where the $i$ refers to a dimension. If we sample from the distribution we cannot perform these derivatives. In variational autoencoders this problem is removed by the "reparameterization trick" (Kingma & Welling, 2013) which pushes the stochasticity to an input node, which does not need to be differentiated through, rather than in the middle of the network. A standard variational autoencoder uses Gaussian distributions. For a univariate Gaussian the trick is to turn the latent variable, $h \sim q(h|x) = \mathcal{N}(\mu, \sigma^2)$, which cannot be differentiated through, into $z = \mu + \sigma\epsilon$ where $\epsilon \sim \mathcal{N}(0, 1)$. The parameters $\mu$ and $\sigma$ are both deterministic and can be differentiated through and the stochasticity is added from outside.

We cannot use the Gaussian distribution because we need our $h_i$ terms to be non-negative to fulfill the requirement of NMF. There are several choices for probability distributions which produce only non-negative samples, we use the Weibull distribution for reasons detailed later in this section. We can sample from the Weibull distribution using its inverse cumulative distribution and the input of a uniform distribution at an input node. In Figure 2 we display this method of imposing stochasticity from an input node through $\epsilon$. Similarly to using a standard autoencoder for performing NMF the same restrictions, such as $\mathbf{W}_f$ being forced to stay non-negative, apply to the PAE-NMF.

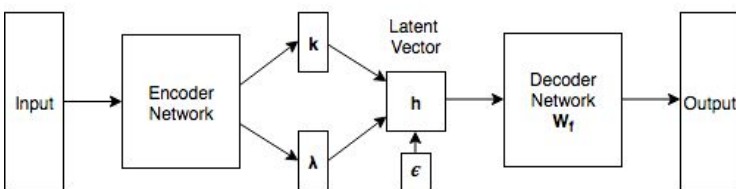

Figure 2: General PAE-NMF with stochasticity provided by the input vector $\epsilon$.

## 2.3 DETAILS OF THE PAE-NMF

In this paper we have utilised the Weibull distribution which has a probability density function (PDF) of

$$f(x) = \begin{cases} \frac{k}{\lambda}\left(\frac{x}{\lambda}\right)^{k-1} \exp\left(-(x/\lambda)^k\right) & \text{if } x \geq 0 \\ 0 & \text{if } x < 0 \end{cases}$$

with parameters $k$ and $\lambda$. The Weibull distribution satisfies our requirements: that the PDF is zero below $x = 0$, falls towards 0 as $x$ becomes large and is flexible enough to enable various shapes of distribution. For each data point there will be $r$ Weibull distributions generated, one for each of the subspace dimensions.

To perform PAE-NMF we need an analytical form of the KL divergence, so we can differentiate the objective function, and a way of extracting samples using some outside form of stochasticity. The KL divergence between two Weibull distribution is given by (Bauckhage, 2014)

$$D_{KL}(F_1||F_2) = \int_0^\infty f_1(x|k_1,\lambda_1) \log\left(\frac{f_1(x|k_1,\lambda_1)}{f_2(x|k_2,\lambda_2)}\right) dx$$

$$= \log\left(\frac{k_1}{\lambda_1^{k_1}}\right) - \log\left(\frac{k_2}{\lambda_2^{k_2}}\right) + (k_1 - k_2)\left[\log(\lambda_1) - \frac{\gamma}{k_1}\right] \qquad (3)$$

$$+ \left(\frac{\lambda_1}{\lambda_2}\right)^{k_2} \Gamma\left(\frac{k_2}{k_1} + 1\right) - 1$$

where $\gamma \approx 0.5772$ is the Euler-Mascheroni constant and $\Gamma$ is the gamma function.

In other situations using NMF with probability distributions the gamma distribution has been used (Squires et al., 2017). The reason that we have chosen the Weibull distribution is that while it is possible to apply variations on the reparameterization trick to the gamma function (Figurnov et al., 2018) it is simpler to use the Weibull distribution and use inverse transform sampling. To sample from the Weibull distribution all we need is the inverse cumulative function, $C^{-1}(\epsilon) = \lambda(-\ln(\epsilon))^{1/k}$. We generate a uniform random variable $\epsilon$ at an input node and then sample from the Weibull distribution with $\lambda$ and $k$ by $C^{-1}(\epsilon)$. So, refering to Figure 2, we now have $\epsilon \sim \mathcal{U}(0,1)$ and each of the dimensions of $\mathbf{z}$ are found by $z_i = C_i^{-1}(\epsilon_i)$.

It is worth considering exactly how and where PAE-NMF differs from standard NMF. In Figure 1 we see that, in terms of NMF, it is fairly unimportant what occurs before the constriction layer in terms of the outputs of interest ($\mathbf{W}$ and $\mathbf{H}$). The aim of the design before that part is to allow the network to find the best possible representation that gets the lowest value of the objective function. There are effectively two parts which make this a probabilistic method: the choice of objective function and the fact that we sample from the distribution. Without those two parts the link between the distribution parameters and $\mathbf{h}$ would just be a non-linear function which might do better or worse than any other choice but would not make this a probabilistic method.

## 2.4 Methodology

We now have the structure of our network in Figure 2 and the distribution (Weibull) that we are using. The basic flow through the network with one hidden layer, for an input datapoint $\mathbf{v}$, looks like:

$$\boldsymbol{\lambda} = f(\mathbf{W}_{\boldsymbol{\lambda}}\mathbf{v}), \quad \boldsymbol{k} = g(\mathbf{W}_{\mathbf{k}}\mathbf{v}), \quad \mathbf{h} = C_{\boldsymbol{\lambda},\mathbf{k}}^{-1}(\boldsymbol{\epsilon}), \quad \hat{\mathbf{v}} = \mathbf{W}_f\mathbf{h} \qquad (4)$$

where $f$ and $g$ are non-linear functions that work element-wise. The inverse cumulative function, $C_{\boldsymbol{\lambda},\mathbf{k}}^{-1}$, also works element-wise.

There are a range of choices to make for this network, to demonstrate the value of our technique we have kept the choices as simple as possible. We use rectified linear units (ReLU) as the activation function as these are probably the most popular current method (Nair & Hinton, 2010). To keep the network simple we have only used one hidden layer. We update the weights and biases using gradient descent. We keep the $\mathbf{W}_f$ values non-negative by setting any negative terms to zero after updating the weights. We did experiment with using multiplicative updates (Lee & Seung, 1999) for the $\mathbf{W}_f$ weights but found no clear improvement. We use the whole data-set in each batch, the same as is used in standard NMF. We use a prior of $k = 1$ and $\lambda = 1$. We calculate the subspace size individually for each data-set using the method of Squires et al. (2017). The learning rates are chosen by trial and error.

To demonstrate the use of our technique we have tested it on three heterogeneous data-sets, displayed in Table 1. The faces data-set, are a group of $19 \times 19$ grey-scale images of faces. The Genes data-set is the 5000 gene expressions of 38 samples from a leukaemia data-set and the FTSE 100 data is the share price of 94 different companies over 1305 days.

Table 1: Data-sets names, the type of data, the number of dimensions, m, number of data-points, n and the source of the data.

| Name | Type | $m$ | $n$ | Source |
|------|------|-----|-----|--------|
| Faces | Image | 361 | 2429 | http://cbcl.mit.edu/software-datasets /FaceData2.html |
| Genes | Biological | 5000 | 38 | http://www.broadinstitute.org/cgi-bin /cancer/datasets.cgi |
| FTSE 100 | Financial | 1305 | 94 | Bloomberg information terminal |

## 3 RESULTS AND DISCUSSION

First we demonstrate that PAE-NMF will produce a reasonable recreation of the original data. We would expect the accuracy of the reconstruction to be worse than for standard NMF because we impose the KL divergence term and we sample from the distribution rather than taking the latent outputs deterministically. These two impositions should help to reduce overfitting which results in the higher reconstruction error.

In Figure 3 we show recreations of the original data for the three data-sets from Table 1. The faces plots show five original images chosen at random above the recreated versions (with $r = 81$). The bottom left plot shows nine stocks from the FTSE 100 with the original data as a black dashed line and the recreated results as a red dotted line ($r = 9$). The results here follow the trend well and appear to be ignoring some of the noise in the data. In the bottom right plot we show 1000 elements of the recreated matrix versus the equivalent elements ($r = 3$). There is significant noise in this data, but there is also a clear trend. The black line shows what the results would be if they were recreated perfectly.

Secondly, we want to look at the $\mathbf{W}_f$ matrices (the weights of the final layer). In NMF the columns of $\mathbf{W}$ represent the dimensions of the new subspace we are projecting onto. In many circumstances we hope these will produce interpretable results, which is one of the key features of NMF. In Figure 4 we demonstrate the $\mathbf{W}_f$ matrices for the faces data-set, where each of the 81 columns of $\mathbf{W}_f$ has been converted into a $19 \times 19$ image (left) and the FTSE 100 data-set (right) where the nine columns are shown. We can see that the weights of the final layer does do what we expect in that these results are similar to those found using standard NMF (Lee & Seung, 1999). The faces data-set produces a representation which can be considered to be parts of a face. The FTSE 100 data-set can be viewed as showing certain trends in the stock market.

We now want to consider empirically the effect of the sampling and KL divergence term. In Figure 5 we show the distributions of the latent space of one randomly chosen datapoint from the faces data-set. We use $r = 9$ so that the distributions are easier to inspect. The left and right set of plots show results with and without the KL divergence term respectively. The black and blue dashed lines show samples extracted deterministically from the median of the distributions and sampled from the distribution respectively. The inclusion of the KL divergence term has several effects. First, it reduces the scale of the distributions so that the values assigned to the distributions are significantly lower in the left hand plot. This has the effect of making the distributions closer together in space. The imposition of randomness into the PAE-NMF through the uniform random variable $\epsilon$ has the effect of reducing the variance of the distributions. When there is a KL divergence term we can see that the distributions follow fairly close to the prior. However, once the stochasticity is added this is no longer tenable as the wide spread of data we are sampling from becomes harder to learn effectively from. When there is no KL divergence term the effect is to tighten up the distribution so that the spread is as small as possible. This then means we are approaching a point, which in fact would return us towards standard NMF. The requirement for both the KL divergence term and the stochasticity means that we do get a proper distribution and the results are prevented from simply reducing towards a standard NMF formulation.

A potentially valuable effect of PAE-NMF is that we would expect to see similar distributions for similar data-points. In Figure 6 we can see the distributions of the $\mathbf{h}$ vectors for two pairs of faces, each pair is similar to the other one in the pair and very different from the other pair. Next to them we plot the distributions. The distributions for the two faces on the left are the black dashed lines

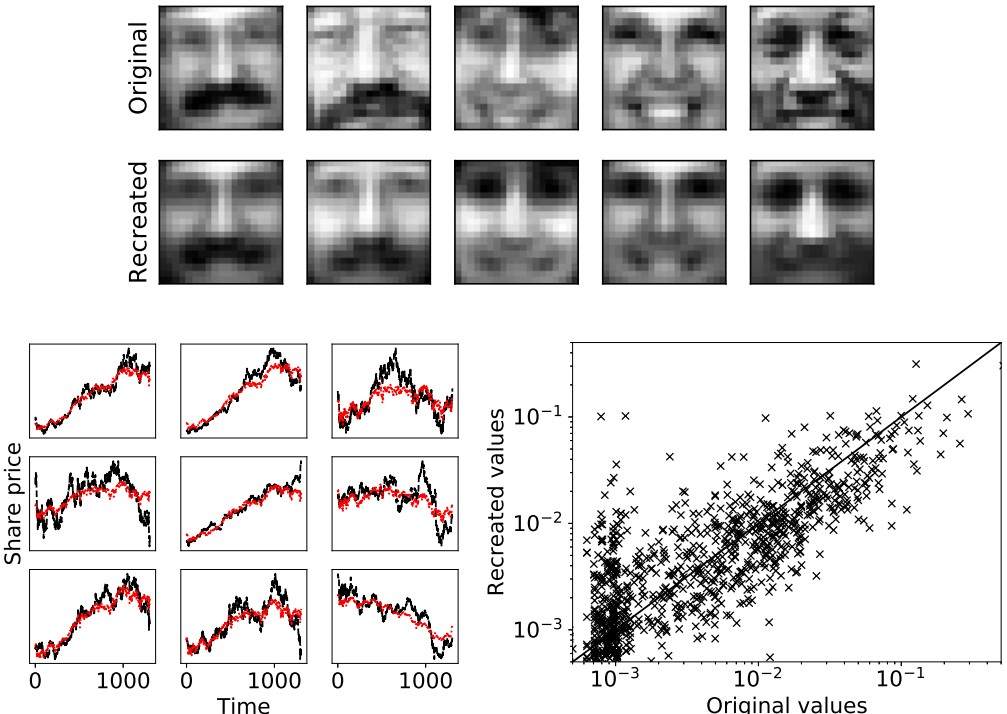

Figure 3: (Top) Top five are original faces with the equivalent recreated faces below. (Bottom left) Nine stocks with original values (black dashed) and recreated values (red dotted). (Bottom right) 1000 recreated elements plotted against the equivalent original.

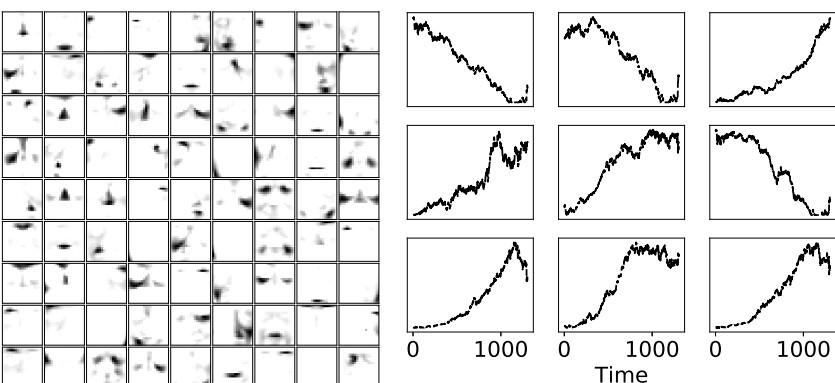

Figure 4: (Left) Each small image is one of the 81 reshaped columns of $\mathbf{W}_f$ for the faces data-set. The features we see are very similar to what you get in standard NMF. (Right) Each plot is a column of $\mathbf{W}_f$ for the FTSE 100 data-set.

and the distributions for the two faces on the right are given by the red dotted lines. There is very clear similarity in distribution between the similar faces, and significant differences between the dissimilar pairs.

Finally, we want to discuss the generation of new data using this model. We show results for the faces and FTSE 100 data-sets in Figure 7. For the faces we use a low $r = 9$ value and show four random faces (left column), with the median being the $\mathbf{h}$ drawn from the middle of the inverse cumulative

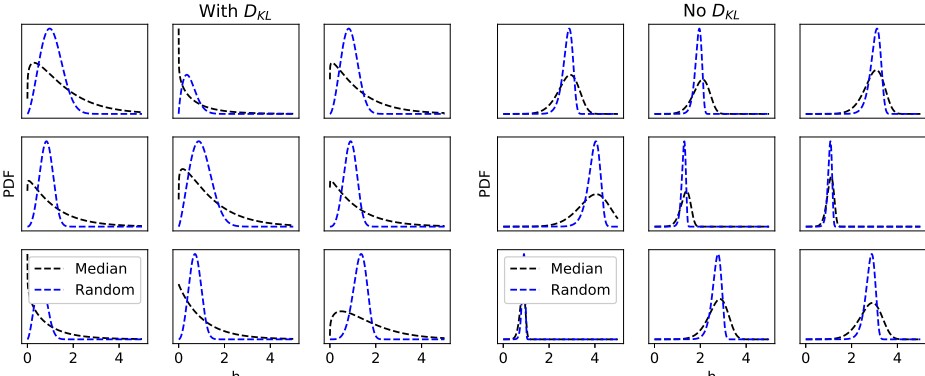

Figure 5: The distributions of $\mathbf{h}$ for one data-point from the faces data-set with $r = 9$. (Left) These plots show the distributions when we include the $D_{KL}$ term. (Right) The $D_{KL}$ term is not applied during training. The black dashed line shows results when we train the network deterministically using the median value of the distribution and the blue line is when we trained with random samples.

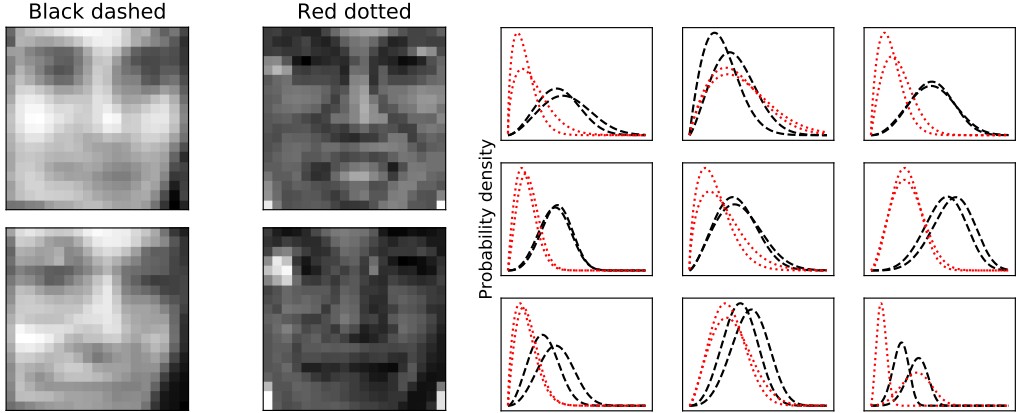

Figure 6: The left images (top and bottom) are similar to one another and we plot their distributions as black dashed lines in the plots to the right. The right images are very different to the left, we plot their distributions as red dotted lines. The similar images have very similar distributions for these data-points.

distribution. The three image plots on the right are then drawn randomly from the distributions. Four stocks from the FTSE 100 data are shown on the right of the figure. The solid lines are the original data with the dotted lines representing sampled data. This ability to directly sample from the distributions and produce plausible new data-points is one of the most useful features of using PAE-NMF over standard NMF.

## 4 SUMMARY

We have demonstrated a novel method of probabilistic NMF using a variational autoencoder. This model extends NMF by providing us with uncertainties on our $\mathbf{h}$ vectors, a principled way of providing regularisation and allowing us to sample new data. The advantage over a VAE is that we should see improved interpretability due to the sparse and parts based nature of the representation formed, especially in that we can interpret the $\mathbf{W}_f$ layer as the dimensions of the projected subspace.

Our method extracts the useful information from the data without over-fitting due to the combination of the log-probability with the KL divergence. While the log-probability works to minimise the error,

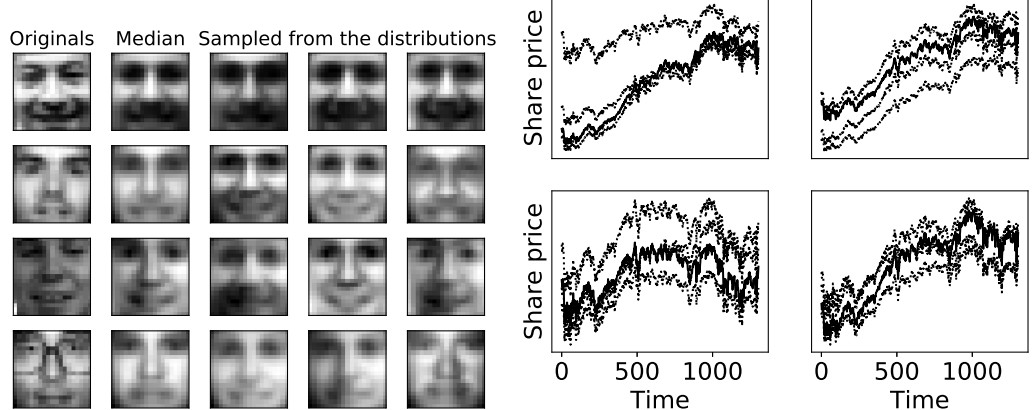

Figure 7: (Left) Sampling from the distributions of the faces data-set with $r = 9$. Four original faces are on the left column, with faces drawn deterministically from the centre of the distribution next to them and three sampled faces along the next three columns. (Right) Sampling from the FTSE 100 data-set with $r = 9$ for four different stocks. The solid black line is the real data and the dotted lines show three sampled versions.

the KL divergence term acts to prevent the model over-fitting by forcing the distribution $q_\phi(\mathbf{h}|\mathbf{x})$ to remain close to its prior distribution.

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
