# OpenReview forum: "A Variational Autoencoder for Probabilistic Non-Negative Matrix Factorisation"
_ICLR.cc/2019/Conference_

### Official Review · AnonReviewer2 · 2018-10-15
**Well written, interesting new idea, modest technical contribution, limited demonstration.**

**Rating:** 7
**Confidence:** 5

**Review:**

TITLE
A VARIATIONAL AUTOENCODER FOR PROBABILISTIC NON-NEGATIVE MATRIX FACTORISATION

REVIEW SUMMARY

Well written, interesting new idea, modest technical contribution, limited demonstration.

PAPER SUMMARY

The paper presents an approach to NMF within a variational autoencoder framework. It uses a Weibull distribution in the latent space.

QUALITY

The work appears technically sound except for minor typos.

CLARITY

Overall the paper is a pleasure to read. Only the presentation of the standard vae could be more clear.

ORIGINALITY

The method is (to my knowledge) novel.

SIGNIFICANCE

I think this paper is a significant contribution. I feel I have learned something from reading it, and am motivated to try out this approach. I believe there should be a wide general interest. The technical contribution is perhaps somewhat modest, as the paper fairly straightforwardly includes non-negativity in a vae setting, but I think this is a good idea. The demonstration of the algorithm is also quite limited - I would have enjoyed seeing this applied to some more reaslistic, practical problems, where perhaps the quantification of uncertaincy (which is one of the main benefits of a vae-based nmf) would come more directly into play.

FURTHER COMMENTS

page 3

The presentation of the VAE objective is a bit oblique. The statement "they require a different objectiv function" is not wrong, but also not very precise. The equality in eq. (2) is incorrect (I assume this is meant to be a stochastic approximation, i.e. the expectation over q approximated by sampling?)

"with \hat v  the reconstructed vector" Not clear. I assume \hat v is reconstructed from a sample from q given v ?

There is a typo in eq. 3. The first factor in the second to last term should be (lambda_1/\lambda_2)^(k_2)

---

> ### Public Comment · (anonymous) · 2018-11-19
> **Author response to reviewer**
>
> We thank this reviewer for their generous review.
>
> Comment: The presentation of the VAE objective is a bit oblique. The statement "they require a different objectiv function" is not wrong, but also not very precise. The equality in eq. (2) is incorrect (I assume this is meant to be a stochastic approximation, i.e. the expectation over q approximated by sampling?)
> Our response: the first point here is very valid – that is an imprecise sentence and we will improve it. The second point is also correct,  we should not have had an equality there as it is an approximation, we will also fix that in the manuscript.
>
> Comment: "with \hat v the reconstructed vector" Not clear. I assume \hat v is reconstructed from a sample from q given v ?
> Our response: yes that is correct, we will add a sentence about how vhat is formed there.
>
> Comment: There is a typo in eq. 3. The first factor in the second to last term should be (lambda_1/\lambda_2)^(k_2)
> Our response: Thank you very much for spotting that typo, we will correct it.

---

> > ### Comment · AnonReviewer2 · 2018-12-04
> > **Thank you**
> >
> > Thank you for the clarification

---

### Official Review · AnonReviewer3 · 2018-10-24
**Experimental evaluations are mostly qualitative**

**Rating:** 4
**Confidence:** 3

**Review:**

This paper replaces the Gaussian latent variables in standard VAE with the Weibull distribution and therefore presents a VAE solution to nonnegative matrix factorization, under squared Euclidean distance loss function. The literature review summarizes four related work very well. The adopted Weibull distribution provides a tractable inverse CDF function and analytical form of the KL divergence, facilitating VAE inference. In particular, the effects of the entropy term are discussed in detail.  Experiments illustrate the generated data from the model,  the learned part-based dictionaries, and the distributions of latent variables from similar data points.

Questions:

1. What is the updating rule for W_f? Is it multiplicative?  In Sec 2.4, The value of W_f is kept to be nonnegative by "setting negative terms to zero". Does it mean once one entry is set to zero, it would never be positive in the sequential gradient steps?

2. Although the proposed model is claimed to be probabilistic, the L2 loss function in equation (2) implies that the data generated from the model could be negative. How would the proposed approach handle other loss function of NMF such as KL (e.g., under Poisson assumption)?

3. The nonnegative variant sounds interesting, but the experimental results are quite limited. It is unclear how the proposed approach would compare to other probabilistic NMF models and algorithms, or the standard VAE as a generative model. It seems the proposed method can do as good as NMF or VAE in some aspects. This begs the question of when would the proposed approach be superior to others and in what aspect?

Minor:
In some places where the parameter are constrained to be nonnegative, it would be more clear to use notations such as R_+ instead of R.

---

> ### Public Comment · (anonymous) · 2018-11-19
> **Author response to reviewer**
>
> Thank you very much for the review. We will answer each comment in turn.
>
> Comment 1. What is the updating rule for W_f? Is it multiplicative? In Sec 2.4, The value of W_f is kept to be nonnegative by "setting negative terms to zero". Does it mean once one entry is set to zero, it would never be positive in the sequential gradient steps?
> Our response: we use a gradient descent method, not a multiplicative update rule which means that values of Wf which have gone to zero can become non-zero again. We did not discuss this in our paper as it is standard in neural networks to use gradient descent (or variants of) methods, while in NMF methods such as Lee and Seung’s multiplicative update is commonly used. We will add a brief mention of the update method in our resubmitted paper.
>
> Comment 2. Although the proposed model is claimed to be probabilistic, the L2 loss function in equation (2) implies that the data generated from the model could be negative. How would the proposed approach handle other loss function of NMF such as KL (e.g., under Poisson assumption)?
> Our response: we are a little unclear about the first part of this comment. The data, v-hat, generated by the model cannot be negative because the output v-hat=Wf*h where h is draw from the Weibull distribution, which is non-negative, and Wf is forced to remain non-negative by the algorithm. It is certainly true that the difference between v-hat and v can be positive or negative. We have not investigated the use of the KL divergence for the error term but we see no obvious reason why this would not work effectively.
>
> Comment 3. The nonnegative variant sounds interesting, but the experimental results are quite limited. It is unclear how the proposed approach would compare to other probabilistic NMF models and algorithms, or the standard VAE as a generative model. It seems the proposed method can do as good as NMF or VAE in some aspects. This begs the question of when would the proposed approach be superior to others and in what aspect?
> Our response: here we refer to our reply to Comment 2 from the first reviewer and copy the same comment below. The experiments are there to back up the idea which is the main part of the paper. It is challenging in many unsupervised learning problems to conclusively show the technique is effective. In this case we were trying to demonstrate some key features of our method.
>
> Although probabilistic methods for NMF have been developed, even a full Bayesian framework still faces the problem that, for the vast majority of problems where NMF is used, we have little idea about what is the appropriate prior. We would therefore be forced to do model selection or introduce hyperparameters and perform an inference (maximum likelihood or Bayesian) based on the evidence. However, as the posterior is such cases is unlikely to be analytic this is likely to involve highly time consuming Monte Carlo. In doing so we would expect to get results close to those we obtain. However, for machine learning algorithms to be of value they must be practical. Our approach, following a minimum description length methodology, provides a principled method for achieving automatic regularisation. Because it fits within the framework of deep learning it is relatively straightforward and quick to implement (using software such as Keras or pytorch with built-in automatic differentiation, fast gradient descent algorithms, and GPU support). In addition our approach provides a considerable degree of flexibility (e.g. in continuous updating, allowing exogenous data, etc.), which we believe might be much more complicated to achieve in a fully probabilistic approach. Obviously we failed to properly articulate these advantages in our original submission and will correct this when we submit our corrections.
>
> Minor Comments:
> In some places where the parameter are constrained to be nonnegative, it would be more clear to use notations such as R_+ instead of R.
> Our response: a fair point which could add to clarity – we will make this change in our manuscript.

---

### Official Review · AnonReviewer1 · 2018-10-29
**lacking motivation and comparisons**

**Rating:** 4
**Confidence:** 5

**Review:**

The paper is generally well-written (lacking details in some sections though). My main criticism is about the lack of motivation for nonnegative VAE and lack of comparison with NMF.

Comments:
- the motivation of the proposed methodology is not clear to me. What is the interest of the proposed auto-encoding strategy w.r.t NMF ? There is no experimental comparison either. Besides the probabilistic embedding (which exists in NMF as well), is there something PAE-NMF can do better than NMF ? There is probably something, but the paper does not bring a convincing answer.
- the paper missed important references to nonnegative auto-encoders, in particular:
https://paris.cs.illinois.edu/pubs/paris-icassp2017.pdf
- the review of probabilistic NMF works is limited, see e.g.,
https://paris.cs.illinois.edu/pubs/smaragdis-spm2014.pdf
- more details are needed about inference in Section 2.4

Minor comments:
- the notations z and h are sometimes confusing, what about using h every where ?
- it’s not clear to me how the first term in (1) is equal to the second term in (2

---

> ### Public Comment · (anonymous) · 2018-11-19
> **Author reply to reviewer**
>
> Thank you very much for the review. We will answer each comment in turn.
>
> Comment 1: the motivation of the proposed methodology is not clear to me. What is the interest of the proposed auto-encoding strategy w.r.t NMF ? There is no experimental comparison either. Besides the probabilistic embedding (which exists in NMF as well), is there something PAE-NMF can do better than NMF ? There is probably something, but the paper does not bring a convincing answer.
> Our response: Although probabilistic methods for NMF have been developed, even a full Bayesian framework still faces the problem that, for the vast majority of problems where NMF is used, we have little idea about what is the appropriate prior. We would therefore be forced to do model selection or introduce hyperparameters and perform an inference (maximum likelihood or Bayesian) based on the evidence. However, as the posterior is such cases is unlikely to be analytic this is likely to involve highly time consuming Monte Carlo. In doing so we would expect to get results close to those we obtain. However, for machine learning algorithms to be of value they must be practical. Our approach, following a minimum description length methodology, provides a principled method for achieving automatic regularisation. Because it fits within the framework of deep learning it is relatively straightforward and quick to implement (using software such as Keras or pytorch with built-in automatic differentiation, fast gradient descent algorithms, and GPU support). In addition our approach provides a considerable degree of flexibility (e.g. in continuous updating, allowing exogenous data, etc.), which we believe might be much more complicated to achieve in a fully probabilistic approach. Obviously we failed to properly articulate these advantages in our original submission and will correct this when we submit our corrections.
>
> In terms of experimental comparison – that is always challenging in unsupervised learning, especially with limited space to go deeply into the comparison. We could add equivalent NMF results to some of the figures we provide, however, we do not think it would add significantly to the purpose of the paper which is to introduce the method and idea behind PAE-NMF.
>
> Comment 2: the paper missed important references to nonnegative auto-encoders, in particular:
> https://paris.cs.illinois.edu/pubs/paris-icassp2017.pdf (https://paris.cs.illinois.edu/pubs/paris-icassp2017.pdf)
> Our response: This is a very fair point, we are happy to add in this reference to this paper.
>
> Comment3 : the review of probabilistic NMF works is limited, see e.g.,
> https://paris.cs.illinois.edu/pubs/smaragdis-spm2014.pdf
> Our response: We are happy to extend our review of probabilistic NMF in this paper but should note that we are at the limit of 8 pages at the moment and there will always be a limitation on what can be added.
>
> Comment:4  more details are needed about inference in Section 2.4
> Our response: we certainly could write considerably more but as our method is based upon the work of VAEs we did not want to go over work that has already been done and discussed. Also, we are at the edge of the page limit.
>
> Minor comments:
> Minor Comment: the notations z and h are sometimes confusing, what about using h every where ?
> Our response: The use of z was due to its familiarity to researchers with a background in VAEs. If this was confusing we are happy to make this change whilst altering our paper.
>
> Minor Comment: it’s not clear to me how the first term in (1) is equal to the second term in (2
> Our response: this should have been an approximation sign rather than an equals sign. We are assuming (not an unreasonable assumption) that the errors, the gap between v-hat and v, will be approximately Gaussian. This is a common assumption made in nearly all VAEs.

---

### Meta-Review · Area_Chair1 · 2018-12-13
**Somewhat incremental and Missing NMF baselines**

**Confidence:** 4
**Recommendation:** Reject

**Metareview:**

The paper introduces a variant of the variational autoencoder (VAE) for probabilistic non-negative matrix factorization. The main idea is to use a Weibull distribution in the latent space. There is agreement among the reviewers that the paper is technically sound and well written, but that it lacks in motivation and demonstration of utility of the proposed method.
All the reviewers think the approach is not particularly novel and somewhat incremental. The main issue is that the empirical evaluation of the algorithm is also quite limited. Specifically, it should have been compared with Bayesian NMF. Many papers have addressed Bayesian NMF with variational inference (Cemgil; Fevotte & Dikmen; Hoffman, Blei & Cook) like in VAE. Experimentally, Bayesian NMF and the proposed PAE-NMF could easily be quantitatively compared on matrix completion tasks. Overall, there was consensus among the reviewers that the paper is not ready for publication.